# Primary Catheter-Directed Thrombolysis for Porto-Mesenteric Venous Thrombosis (PMVT) in Non-Cirrhotic Patients

**DOI:** 10.3390/jcm11164721

**Published:** 2022-08-12

**Authors:** Chia-Ling Chiang, Huei-Lung Liang, Wen-Chi Chen, Ming-Feng Li

**Affiliations:** 1Department of Radiology, Kaohsiung Veterans General Hospital, Kaohsiung 813, Taiwan; 2Department of Medical Imaging and Radiology, Shu-Zen Junior College of Medicine and Management, Kaohsiung 821, Taiwan; 3Division of Gastroenterology and Hepatology, Department of Internal Medicine, Kaohsiung Veterans General Hospital, Kaohsiung 813, Taiwan

**Keywords:** catheter directed thrombolysis (CDT), porto-mesenteric venous thrombosis (PMVT), percutaneous transhepatic route, non-cirrhotic patients, portal hypertension, ischemic bowel

## Abstract

**Purpose:** To report our thrombolytic technique, treatment strategy, and clinical outcomes for porto-mesenteric venous thrombosis (PMVT) in non-cirrhotic patients. **Methods:** Sixteen acute or chronic non-cirrhotic PMVT patients (mean age: 48.6 years) with imminent intestinal ischemia were enrolled from 2004 to 2020. Eight patients presented thrombus extension into the peripheral mesenteric vein, close to the venous arcade. Transhepatic catheter-directed thrombolysis (CDT) was performed by urokinase infusion (60,000–30,000 IU/h concomitant with heparin 300–400 IU/h), catheter aspiration, and/or balloon dilation/stent placement. Additional intra-arterial mesenteric infusion of urokinase (30,000 IU/h) was given in patients with the peripheral mesenteric venules involved. Transjugular intrahepatic porto-systemic shunt (TIPS) was created in patients with poor recanalization of the intrahepatic portal flow (PV). **Results:** The transhepatic route was adopted in all patients, with adjunct indirect mesenteric arterial thrombolytic infusion in eight patients. A total of up to 20.4 million IU urokinase was infused for 1–21 days’ treatment duration. TIPS was created in three patients with recanalization failure of the intrahepatic PV. Technical success was achieved in 100% of patients with complete recanalization of 80% and partial recanalization of 20%. No major procedure-related complications were encountered. The 30-day mortality rate was 6.7%. The overall two-year primary patency was 84.6%. **Conclusions:** CDT can be performed as a primary salvage treatment once the diagnosis is made. CDT via the transhepatic route with tailored thrombolytic regimen is safe and effective for both acute and chronic PMVT. TIPS creation can be preserved in non-cirrhotic PMVT patients if intrahepatic PV recanalization fails.

## 1. Introduction

Porto-mesenteric venous thrombosis (PMVT) is an uncommon cause of mesenteric ischemia, accounting for only 5–15% of cases [1]. Its outcome is unpredictable, reflecting differences in clot distribution, clot volume, speed of onset, and the presence of the radiological features of intestinal ischemia. Søgaard et al. reported that the 30-day mortality risk was particularly high for patients with mesenteric vein (MV) thrombosis (63.1%) when compared with that of portal vein (PV) thrombosis (15.6%) [2]. The aim of treatment is to reverse or prevent the advancement of diffuse thrombosis in the porto-mesenteric veins (PMV) and to treat its complications [3]. In a recent study, Levigard et al. reported that a median of 39.8% increase in liver volume, improvement of liver function, and platelets count could be achieved after recanalization of even chronic PV thrombosis [4]. Although there are clear guidelines for the use of anticoagulation in the non-cirrhotic acute PMVT [5,6], anticoagulation alone does not necessarily result in spontaneous clot lysis and leaves patients at risk of complications, including intestinal infarction and portal hypertension [7].

Several studies with small patient cohorts attempting the prompt restoration of flow in non-cirrhotic PMVT patients using catheter-directed thrombolysis (CDT), either alone or with the aid of thrombectomy devices via the transhepatic or transjugular routes, have been reported in the literature [7,8,9,10,11,12,13,14,15,16]. Nowadays, the transjugular route is advocated for this purpose because of higher reported bleeding complications (50–60%) of the transhepatic route [9,15]. However, the rationale of creating a transjugular porto-systemic shunt (TIPS) in non-cirrhotic PMVT patients is still a matter of on-going debate. Benmassaoud et al. recently proposed a stepwise thrombolysis regimen, applying local CDT after an initial systemic thrombolysis (ST) treatment failure (63%) [7]. The earlier CDT is started, the higher the recanalization rate and the better the clinical outcomes; therefore, we report our treatment strategy, CDT technique, and clinical outcomes in managing non-cirrhotic PMVT.

## 2. Patients and Methods

We retrieved the medical records of patients with porto-mesenteric venous thrombosis (PMVT) who received percutaneous interventional therapy in Kaohsiung Veterans General Hospital between 2004 and 2020 from its Picture Archiving and Communication System (PACS). The inclusion criteria were: 1. PMVT patients without liver cirrhosis, diagnosed both by contrast enhanced computed tomography/magnetic resonance imaging and laboratory liver function test, and 2. patients with obvious clinical symptoms of variceal bleeding and/or imminent intestinal ischemia. Patients were excluded if they had had major surgery within the preceding 2 weeks, current active bleeding, allergic reaction to urokinase, fibrinogen under 150 mg/dL (normal range: 200–400 mg/dL), or platelet count under 50,000/mL.

This study was conducted in accordance with the Declaration of Helsinki. All patients gave informed consent before the interventional procedure was performed. Informed consent from patients for the retrospective medical review was waived by our Institute of the Review Board.

## 3. Thrombolysis Technique

Under sonographic guidance, transhepatic needle puncture of the intrahepatic PV was performed. A 0.035-inch hydrophilic guidewire (Terumo, Tokyo, Japan) and 4F angio-catheter (RC1, Cordis, CA; J-curve, Terumo) were used to navigate through the occluded venous segments into the distal MV.

For patients with localized PMVT, a 7–10 mm diameter bare metal stent (BMS) was deployed via a 7-F, 25 cm vascular sheath (Terumo, Tokyo) to cover the occluded venous segment, followed by a short-term urokinase infusion (30,000 IU/h), if necessary.

For patients with extensive PMVT, catheter aspiration (7F-Mach I, Boston Scientific, Marlborough, MA, USA) and balloon dilation (6–8 mm in diameter) were performed, then followed by a continuous urokinase infusion (30–60,000 IU/h concomitant with heparin 300–400 IU/h) via an infusion catheter (Multi-Sideport infusion catheter, Cook, Bloomington, IN, USA) in the ordinary ward. Interventional procedures were then repeated every 2–3 days, with BMS (6–10 mm) deployed for any residual stenosis. If the peripheral MV were involved, an intra-arterial urokinase infusion (30,000 IU/h) via the SMA was given for 3–5 days after transhepatic thrombolysis. TIPS creation with deploying a 10 mm BMS was considered in patients with poor recanalization of the intrahepatic portal flow after 7–10 days porto-mesenteric thrombolysis. Continuous CDT was performed until improvement was satisfactory. Intensive monitoring was not necessary during the treatment.

Systemic heparinization (3000 IU) and prophylactic antibiotics (cephalexin, 500 mg) were given after successful catheterization into the distal MV. The patients’ coagulation status was checked before the procedure and every 2–3 days during thrombolysis. If the fibrinogen level ever fell below 150 mg/dL, it was then checked every day. Anticoagulation therapy for at least 6 months was prescribed after discharge. The degree of clot lysis after treatment was divided into partial recanalization (PR) and complete recanalization (CR––defined as more than 90% of thrombus having been removed with brisk flow in the porto-mesenteric veins). Complications were defined as “major” if the duration of hospital stay was extended or treatment was required. Portal vein pressure and portosystemic gradients were not systematically measured in this study.

## 4. Statistical Analysis

Continuous variables were presented as median plus range, or as mean ± 1 standard deviation. Categorical items were expressed as the total value and percentage (%). Kaplan–Meier survival curves were used to measure the patency rates. A *p* value of <0.05 was considered statistically significant. Statistical analysis was done using commercially available statistical software (version 17.0; SPSS, Chicago, IL, USA).

## 5. Results

Sixteen non-cirrhotic patients (male: 12, female: 4) with PMVT from 2004 to 2020 were enrolled in this study (mean age: 48.6 ± 23.2 years old, range: 9–96 years). Nine patients (56.3%) had an identifiable risk factor (myeloproliferative disorder in three, antithrombin III deficiency in two, and antiphospholipid antibody syndrome, protein C deficiency, nephrotic syndrome, and pancreatitis in one patient each). Three patients had occlusions localized at the confluent junction of the main PV and superior mesenteric vein (SMV); five patients had thrombus extension into the proximal intrahepatic portal and mesenteric venous branches; and eight patients had simultaneous thrombus involvement from the peripheral venules of the intrahepatic PV to the peripheral MV close to the venous arcade.

Nine patients had the clinical presentation less than two weeks; three patients less than four weeks; and in the remaining four patients, longer than four weeks. Eight patients developed cavernoma on computed tomography (CT) images. Eight patients had ascites, whereas bowel wall thickening on the CT images was seen in 15 patients and bowel wall thinning in one patient, the last indicating impending bowel ischemia. Interventional treatment was indicated after surgical consultation. Patient demographic data are summarized in Table 1.

The transhepatic right portal approach was adopted in 10 patients and the left portal approach in six patients. Additional indirect thrombolytic infusion via the SMA was performed in eight patients while TIPS was created in three patients.

CR was achieved in all three of the localized PMVT patients with additional small-dose urokinase infusions (0.4 and 1.2 million IU apiece) in two patients. Of the five patients without peripheral MV involvement, CR (Figure 1) was achieved in all, with TIPS creation in one patient (Figure 2). One patient had recurrent gastric variceal bleeding at 25 months after the procedure. She received an endoscopic N-Butyl cyanoacrylate (NBCA) injection with the complication of spilling out the sclerosing agent, resulting in total occlusion of the PMV. The other four patients had patency of their PMVs at their 2–128 months imaging follow-up. As for the eight patients with peripheral MV involvement, one patient refused further CDT on the third day and was shifted to systemic anticoagulant therapy. She died of ischemic bowel complications on the 11th day. This patient was excluded from later analysis. The other seven patients underwent CDT via both transhepatic and mesenteric arterial route. CR was achieved in four patients (Figure 3 and Figure 4) and was free of symptoms for up to 118 months’ follow-up. Three patients had poor recanalization of the intrahepatic portal venules. Of them, TIPS creation was performed in two patients. Although the PMV were angiographically patent in these two patients, the flows were still slow, and so one patient complicated with ischemic bowel and died on the 30th day.

The overall technical success of recanalization in the 15 patients was 100% with CR in 12 patients (80.0%) and PR in three patients (20.0%). The 30-day mortality rate was 6.7% (one patient). Of the three PR patients, one patient died on the 30th day, and the remaining two patients showed re-occlusion of the MPV but with patency of the MV. Of the 12 CR patients, one 95-year-old patient died at 1.5 months after the procedure without imaging follow-up, one patient complicated with total occlusion of the portal vein after her endoscopic NBCA injection, and the other 10 patients presented as free of symptoms for 4–150 months’ clinical follow-up. The treatment techniques, thrombolytic agent dose, and clinical outcomes of these 15 patients are listed in Table 2. Excluding the two patients without follow-up images, the overall two-year primary patency of the PMVT in the 13 patients with 1–128 months (median: 23 months) imaging follow-up was 84.6%.

## 6. Discussion

This study of 16 PMVT patients demonstrated that local CDT with a prolonged, tailored, low-dose, treatment can effectively recanalize extensive PMVT via a transhepatic route and can be used as the primary intervention therapy. Technical success was achieved in 100% of patients with complete recanalization of 80% and partial recanalization of 20%. TIPS was created in three patients with recanalization failure of the intrahepatic PV. No major procedure-related complications were encountered. The 30-day mortality rate was 6.7%. The overall two-year primary patency was 84.6%.

PMVT was reported to have an incidence of 1.8–2.7 persons per 100,000 [17,18]. Thrombosis of the larger portions of the mesenteric vein is mostly secondary to local factors, such as malignancy, pancreatitis, and infection, while thrombosis that originates from the vena recta is most commonly related to a prothrombotic state [3]. Infarction of the bowel mostly requires involvement of the venous arcades and vena recta [18,19].

Current guidelines recommend early anticoagulation for at least six months as the standard treatment for acute PMVT [5,6]. Following early anticoagulation, recanalization occurs in 32–45.4% [7,20]. In the EN-Vie prospective study, the one-year recanalization rates were 39% and 73% for the portal and superior mesenteric veins, respectively [21]. However, complete recanalization was less frequent and was only achieved in 20% of patients (19/95) with cavernoma development in 19.9–40% [20,21]. Furthermore, thrombus resolution is rare in patients with higher degrees of PMVT, ascites, and the presence of more than one prothrombotic risk factor [5,21,22,23]. For such emergencies, successful local CDT has been reported in some small-series studies, either through the transjugular, transhepatic, transsplenic, or a transileocolic venous route [7,8,9,10,13,24,25,26,27]. Liu et al. concluded that indirect thrombolysis via the SMA was safer and more effective than systemic thrombolysis [27]. Another paper reported that the recanalization effect of direct thrombolysis is better than that of indirect treatment (14.3% partial recanalization) [10]. Hollingshead et al. recommended a combined direct and indirect thrombolysis simultaneously for more extensive PMVT patents [9]. In the present study, we started the combination therapy at 3–5 days after PR of PMVT was achieved, in order to avoid the possible bleeding complication due to high concentrations of urokinase retention in the intestinal mucosa, and also to shorten the femoral arterial catheter indwelling time and patients’ discomfort.

Smalberg et al. [15] and Hollingshead et al. [9] reported major-procedure-related bleeding complications in 50% and 60%, respectively, of patients treated with local CDT via the transhepatic approach. Thereafter, the transhepatic approach was regarded as unsafe, and the transjugular approach became the mainstream approach for CDT [24,25,28]. It is worth noting that the thrombolytic dose used in these two studies (100,000–240,000 IU/h urokinase concomitant with 1250 IU/h heparin in Hollingshead’s series [9] and rtPA of 2–4 mg/hour in Smalberg’s series [15]) were around two folds higher than those of others’ and our series [7,10,13]. In Rabuffi’s series [14], they used a 10-Fr atherectomy device via the transhepatic route and followed by thrombolytic therapy without bleeding complication. Accordingly, we concluded that the major bleeding complication of the CDT is more dose-related than the transhepatic route approach.

The technical success rates of TIPS in patients with PMVT were reported to be significantly variable, ranging from 35% to 83% [11,24,25,29,30,31,32,33] with either a transhepatic balloon-assisted technique [32] or by using a thin (21 G) TIPS-set needle with multiple blind punctures under fluoroscopic guidance [24]. It was concluded that TIPS was not recommended in patients with a total fibrotic cord and/or without a large collateral vein or in those with extensive SMV thrombosis [33]. In our series, 100% technical success of the transhepatic approach was achieved, without any procedure-related major bleeding complications, nor were patients excluded due to thrombus extension, fibrotic cord of MPV, or cavernoma formation.

Although current guidelines recommend early anticoagulation as the standard treatment for acute PMVT [5,6], Benmassaoud et al. reported that the recanalization rate and CR from ST could be achieved in only 31.8% and 13.6% of patients, respectively, versus 78.6% and 36.4% in patients receiving local CDT in their series [7], and that longer ST intervals would be associated with a decreased chance of recanalization. Thereafter, they proposed a stepwise protocol using systemic anticoagulation first and then followed by local CDT for treatment-resistant patients [7], even under the circumstance that the thrombus might be chronic. Previous articles, focusing on managing the localized chronic portal venous thrombosis without thrombolysis, reported the technical success of 100% (five patients) via the transhepatic route [5], and 35% and 50% via TIPS [11,25]. To the best of our knowledge, the present study reports the first experience of managing the chronic thrombosis extension into the MV by local CDT without the need of intensive monitoring. The experience of clinical success to dissolve chronic thrombus is similar to a previous report on chronic arterial occlusive disease [34].

Previous studies have shown recanalization rates of CDT for PMVT ranging from 52.3% to 94.1%, with CR of 6.3% to 52.9% [9,13,35,36,37,38] and are summarized in Table 3. Klinger et al. reported that the one- and two-year secondary PV patency rates in 17 consecutive acute non-cirrhotic PMVT patients were both 88.2% without significant differences in patients with TIPS-implantation at the end of thrombolysis, compared to patients without TIPS-implantation (*p* = 0.2) [13]. In the present study, our patients with CR showed good long-term patency. As a result, complete venous recanalization should be the goal of the treatment strategy in these non-cirrhotic PMVT patients. The two-year patency rate in our series was 84.6%.

## 7. Limitations

Firstly, this was a retrospective study with a small patient group that made setting a parallel control group difficult. Unfortunately, this is the reality when studying a rare disease associated with significant morbidity and mortality. Secondly, thrombectomy devices were not used to accelerate the clearance of acute or subacute thrombi in our study, as they are expensive and because of the fact that a clinically acute presentation may actually be caused by a chronic occlusion with acute clinical exacerbation. The real clinical benefits of the thrombectomy devices in these patients may be limited. Lastly, the long-term vascular patency, especially of the partially recanalized PMV, cannot be determined from such a small patient group.

## 8. Conclusions

Acute clinical presentations of PMVT do not reflect the actual thrombus stage. The earlier CDT is started, the higher the recanalization rate and the better the clinical outcomes, and since the transhepatic CDT under ultrasonographic guidance puncture has high technical success and carries few risks with our tailored thrombolytic regimen, it is our recommendation that primary local CDT should be started as the primary therapy once the diagnosis of acute PMVT is made. The goal of treatment strategy for PMVT should be to achieve complete recanalization. CDT via the transhepatic route with a tailored thrombolytic regimen is safe and effective in both acute and chronic PMVT.

## Figures and Tables

**Figure 1 jcm-11-04721-f001:**
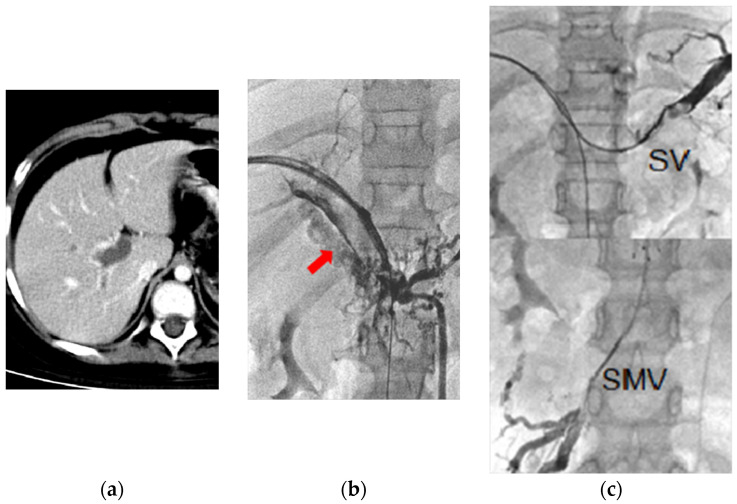
Chronic PMVT in a nine-year-old girl diagnosed with nephrotic syndrome, with complaints of intermittent abdominal pain for six months. (**a**) CT image revealed diffuse PMVT. (**b**) Transhepatic portogram confirmed CT findings. Arrow: cavernoma. (**c**) Upper: splenic vein (SV) thrombosis. Lower: SMV thrombosis. (**d**,**e**) Transhepatic portograms showed recanalization of the SMV, SV, MPV, and intrahepatic portal veins after continuous urokinase infusion (total dose of 6.8 million IU for 12 days’ infusion), BMS placement (8 × 100 mm), and repeated balloon dilation. This patient was symptom-free for more than 10 years at the time of writing, with patent. Vascular flow on the last color Doppler ultrasound follow-up 128 months after the procedure.

**Figure 2 jcm-11-04721-f002:**
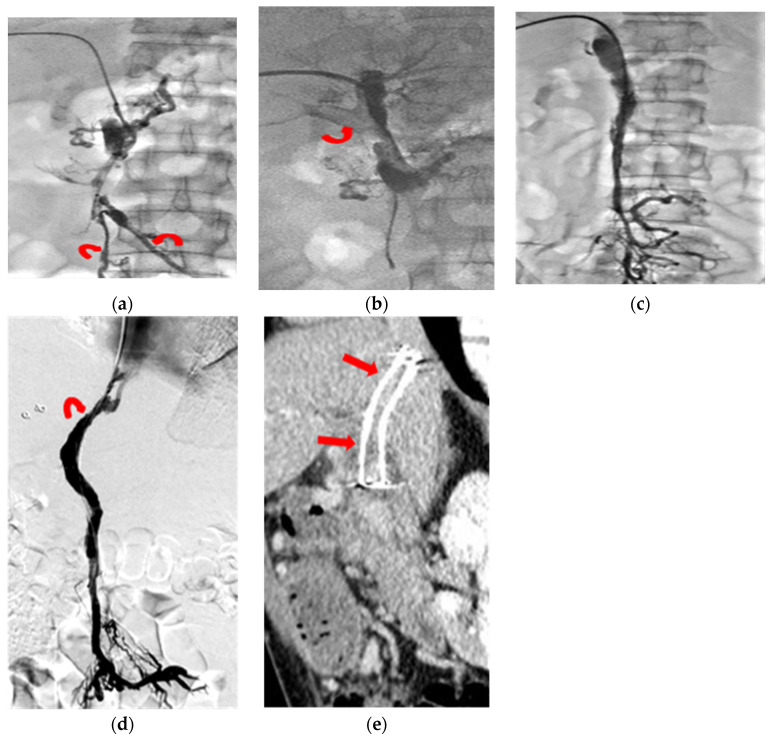
Chronic PMVT with acute exacerbation in a 44-year-old male patient presenting with abdominal pain for 22 days. (**a**) Transhepatic portogram revealed diffuse PMVT involving the peripheral venules of the intra-hepatic PV, leaving the peripheral mesenteric venules spared (curved arrows). (**b**) Transhepatic portogram demonstrated faint partial opacification of the right portal vein (curved arrow) after continuous urokinase infusion and repeated catheter thrombus aspiration. (**c**) Transhepatic portogram showed, although near complete recanalization of the mesenteric veins and the main portal vein, no further thrombus lysis without contrast opacification of the intrahepatic portal thrombus. (**d**) A TIPS shunt (curved arrow) was created on the 10th procedure day to lower the outflow resistance. (**e**) CT images (sagittal oblique reformation) at three months’ follow-up revealed recanalization of the porto-mesenteric vein with patency of the TIPS shunt (arrows).

**Figure 3 jcm-11-04721-f003:**
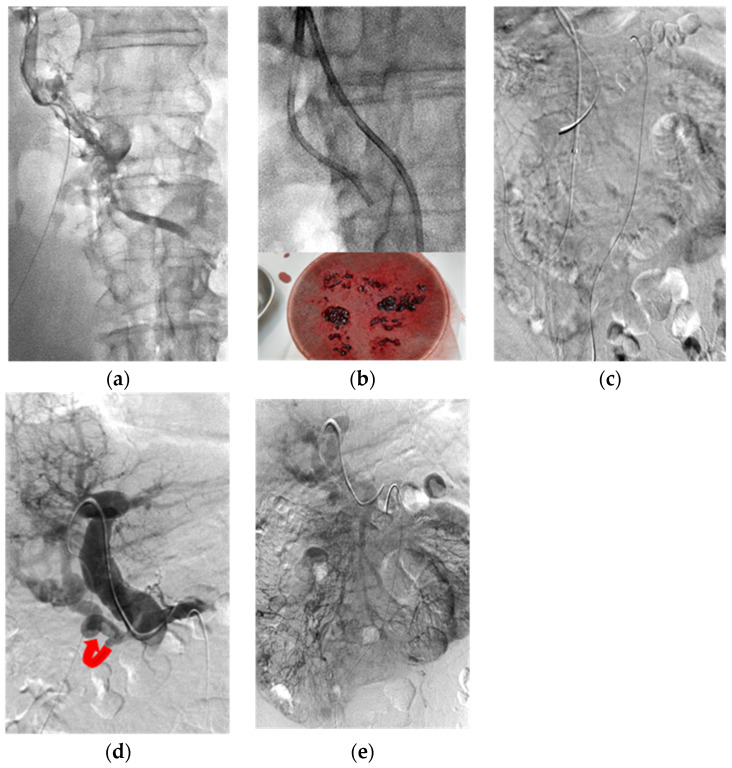
A 58-year-old male patient with abdominal pain for four days. Five years prior, he underwent segmental ileal resection due to venous occlusive ischemic bowel. (**a**) Transhepatic portogram showed extensive PMVT involving the peripheral venules of both the intrahepatic PV and MV close to the venous arcade. (**b**) Upper: Fluoro-image demonstrated catheter aspiration (6F Envoy catheter) in the thrombosed PMV. Lower: photograph showed some fresh and old blood clots aspirated out. (**c**) SMA angiograms (venous phase) failed to opacify the segmental mesenteric veins. (**d**) Transhepatic portogram showed recanalization of the main and intrahepatic PVs after repeated balloon dilation, catheter aspiration and continuous urokinase infusion via both the transhepatic portal (six days’ duration) and transarterial SMA (three days’ duration) routes. Curved arrow: residual cavernoma. (**e**) SMA angiograms (venous phase) demonstrated re-opacification of the mesenteric veins and portal veins after treatments.

**Figure 4 jcm-11-04721-f004:**
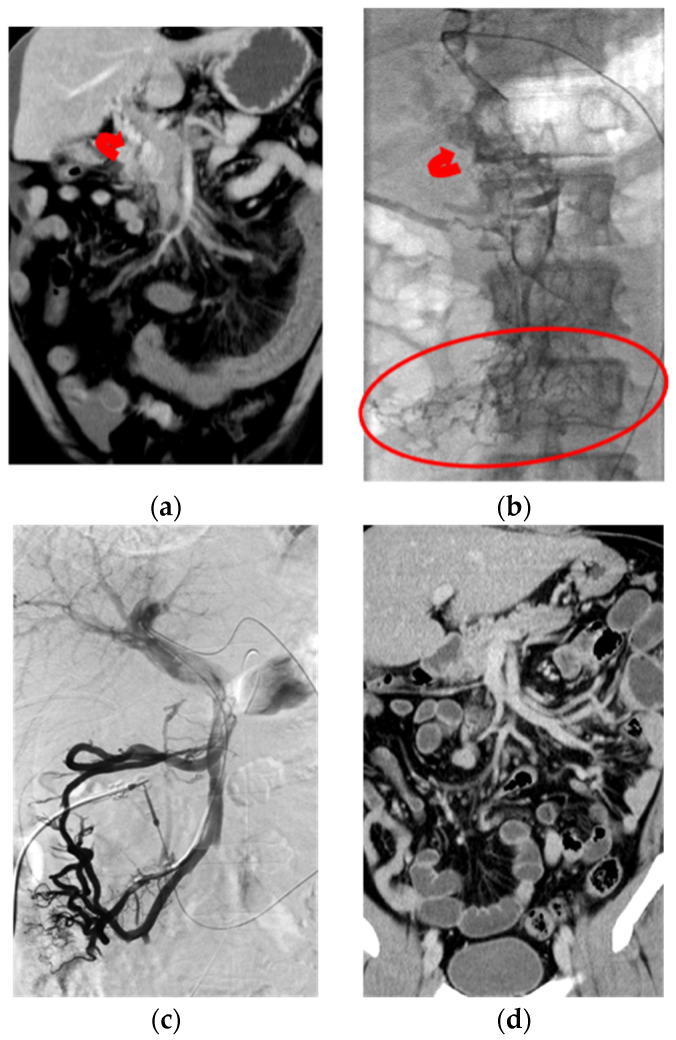
Chronic PMVT with acute exacerbation in a 40-year-old male patient diagnosed as antithrombin III deficiency with complaint of abdominal pain for three days. (**a**) Coronal CT image showed total occlusion of the PMV with bowel wall edema and ascites, as noted on this CT image. There was cavernoma formation (curved arrow) in this acute onset chronic PMVT. (**b**) Transhepatic portogram confirmed the CT findings of diffuse thrombosis involving the peripheral mesenteric venules (circle) and major and intra-hepatic portal veins. Curved arrow: cavernoma formation. (**c**) Transhepatic portogram revealed recanalization of the mesenteric veins and major and intrahepatic portal veins after local CDT, via both transhepatic and SMA routes, with the total urokinase dose of 11.6 million IU for 12 days’ infusion. Non-opacification of the cavernoma was noted on the complete portogram. (**d**) Coronal CT image at eight-month’ follow-up revealed patency of the PMV with normal appearance of the bowel loop.

**Table 1 jcm-11-04721-t001:** Demographic data of the 16 PMVT patients.

No	Sex	Age	Symptoms	Onset	Etiology	A/C	Cavernoma	Wall Edema	Ascites	Peripheral Venules
1	M	69	abdominal pain	3 days	unknown	acute	nil	y	nil	nil
2	F	9	abdominal pain	6 months	nephrotic syndrome	chronic	y	y	y	nil
3	M	37	abdominal pain	7 days	antiphospholipid antibody syndrome	acute	nil	y	nil	PV + MV
4	M	47	abdominal pain	14 days	unknown	acute	nil	y	nil	MV
5	F	11	variceal bleeding	3 years	protein C deficiency	chronic	y	y	y	nil
6	M	26	variceal bleeding	4 months	myeloproliferative disorder	chronic	y	y	nil	nil
7	F	73	abdominal pain	2 months	unknown	chronic	y	y	nil	PV + MV
8	M	96	tarry stool	12 days	unknown	chronic	nil	y	nil	nil
9	M	35	abdominal pain	26 days	myeloproliferative disorder	subacute	y	y	y	PV + MV
10	M	40	abdominal pain	3 days	antithrombin III deficiency	chronic	y	y	y	PV + MV
11	M	76	fever of unknown origin	14 days	antithrombin III deficiency	acute	nil	thinning	nil	PV
12	M	49	abdominal pain	2 days	myeloproliferative disorder	acute	nil	y	y	PV + MV
13	M	59	abdominal pain	4 days	unknown	chronic	y	y	y	PV + MV
14	F	51	abdominal pain	14 days	unknown	subacute	nil	y	y	PV + MV
15	M	56	tarry stool	23 days	pancreatitis	chronic	nil	y	y	nil
16	M	44	abdominal pain	22 days	unknown	chronic	y	y	nil	PV

M: male; F: female; PV: portal vein; MV: mesenteric vein.

**Table 2 jcm-11-04721-t002:** Technical perspectives and clinical outcomes of the 16 PMVT patients.

No	Route	L/R	Lysis Days	UK Dose (Million)	BMS Size (mm)	Recanalization	Follow-Up Image	Status	Survival
1	TH	R	0.4	0.4	7 × 60	CR	CT-1m	patent	4 months-lfu
2	TH	R	12	6.8	8 × 100(2)	CR	US-128m	patent	150 months
3	TH + SMA	L	9	8.6	6 × 120	CR	CT-105m	patent	119 months
4	TH + SMA	L	3	4.5	6 × 60	CR	CT-79m	patent	110 months
5	TH	R	6	2.4	8 × 60	CR	CT-25m	occlusion	106 months
6	TH	R	13	9.6	8 × 100	CR	CT-26m	patent	105 months
7	TH + SMA	R	2	2.8	nil	refused further CDT	nil	excluded	Expired––9 days
8	TH	R	3	0	7 × 60	CR	nil	uncertain	Expired––1.5m
9	TH + SMA + TIPS	R	19	16.4	8 × 80	PR	CT-1m	re-occluded	2 months-lfu
10	TH + SMA	L	12	11.6	nil	CR	CT-65m	patent	68 months
11	TH	L	3	3.6	8 × 37	CR	US-36m	patent	41 months
12	TH + IMA + TIPS	R	12	10.8	10 × 80	PR	nil	uncertain	Expired––30 days
13	TH + SMA	L	6	6.4	nil	CR	MR-15m	patent	25 months
14	TH + SMA	R	21	20.4	8 × 40	PR: refused TIPS	CT-1m	re-occluded	18 months
15	TH	R	1	1.2	10 × 37	CR	CT-5m	patent	14 months
16	TH + TIPS	L	15	12.9	8 × 60	CR	CT-2m	patent	5 months

TH: transhepatic; SMA: superior mesenteric artery; TIPS: transjugular intrahepatic porto-systemic shunt; R: right; L: left; UK: urokinase. BMS: bare-metal stent; CR: complete recanalization; PR: partial recanalization; CDT: catheter-directed thrombolysis; lfu: loss to follow-up.

**Table 3 jcm-11-04721-t003:** Main studies of invasive therapy for non-cirrhotic PMVT patients reported in the literature.

Author	Year	Cirrhosis/Non-C	Route TJ/TH/SMA	Acute/Chronic	Thrombolytic Agent	Thrombectomy Device	Technical Success	CR/PR
Bilbao [8]	2004	0/6	TJ/TH	chronic	nil/bare metal stent	nil	100%	nr
Hollingshead [9]	2005	0/20	TH/SMA	acute	urokinase/rt-PA	C-aspiration	100%	15%/60%
Smalberg [15]	2008	2/10	TH/TJ	acute	rt-PA	nil	100%	25%/33.3%
Liu [10]	2009	14/32	TJ/TH/SMA	acute	urokinase	C-aspiration	100%	56.5%/17.4%
Fanelli [24]	2011	0/12	TJ	chronic	nil/stentgraft	nil	83.3%	90%/10%
Qi [25]	2012	0/20	TJ	chronic	nil/bare metal stent	nil	35%	nr
Luo [11]	2014	0/15	TJ	chronic	nil/stentgraft	nil	73.3%	nr
Song [16]	2017	0/8	TH	acute	urokinase	yes	100%	37.5%/62.5%
Klinger [13]	2017	0/17	TJ	acute	Urokinase/rt-PA	C-aspiration	94.1%	56.3%/43.8%
Rosenqvist [12]	2018	2/4	TJ	acute	alteplase	yes	100%	83.3%/16.7%
Rabuffi [14]	2020	1/7	TJ/TH	acute	urokinase	yes	100%	nr
Benmassaoud [7]	2019	0/14	TJ	acute	rt-PA	yes	78.6%	36.4%/45.4%
current study	2022	0/16	TH/SMA/TJ	acute/chronic	urokinase	C-aspiration	100%	80%/20%

Non-C: non-cirrhosis; C-aspiration: catheter aspiration; TJ: transjugular; TH: transhepatic; SMA: superior mesenteric artery; CR: complete recanalization; PR: partial recanalization; nr: not reported.

## Data Availability

The authors declare that this report does not contain any personal information that could lead to the identification of the patient(s). The authors declare that they obtained a written informed consent from the patients and/or volunteers included in the article. The authors also confirm that the personal details of the patients and/or volunteers have been removed.

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
