# Peer review of "Primary Catheter-Directed Thrombolysis for Porto-Mesenteric Venous Thrombosis (PMVT) in Non-Cirrhotic Patients"

_jcm, 2022, doi:10.3390/jcm11164721_

Round 1

Reviewer 1 Report

I read with interest this article concerning reports about thrombolytic technique in porto-mesenteric thromboses.

It concerns an expert method, and the literature is rather poor. As it concerns a retrospectice study, with a small population, the texte must be improved before any publication.

- Patients characterisitcs are better in results report.

- Discussion paragraph must be improved: the text classical order is not usual (In first, summarize th results.).

- It would be interesting to havec, if possible, a comparison with a group without invasice technic. If not, it will be explained.

- A table with the main studies with invasive techincs should be interesting. In fine, place of this technique in acute portal thrombosis must be discussed: would it change the recomendations?

Author Response

Dear Editor of of Journal of Clinical Medicine,

Thank you for the opportunity to revise and resubmit our manuscript entitled “Primary catheter-directed thrombolysis for porto-mesenteric venous thrombosis (PMVT) in non-cirrhotic patients.We found the reviewers’ comments to be helpful and have carefully responded to each suggestion.

We have included a response to the reviewers in which we address each comment. In our response, the reviewers’ comments are in black text, and our responses follow below, in blue, and are prefaced by “Reply:” Corresponding changes are also in blue in the manuscript text in the revised file

Thank you for your consideration.

Sincerely,

Huei-Lung Liang, M.D.

Department of Radiology

Kaohsiung Veterans General Hospital, Taiwan

Reviewer 2 Report

This paper evaluated the effectiveness of primary catheter directed thrombolysis (CDT) on porto-mesenteric venous thrombosis (PMVT), in which usually ST is performed primarily. The detail procedures, images, and outcomes were described, and the outcomes were clearly better than those of previous studies. However, there are several points which should be modified, including an emphasis on the strength of the technique the authors applied compared to previous techniques. Lack of the control group is also one of the strong limitations.

Major concerns and questions for authors:

1.     The most significant concern is that, there is significant variability in the results of ST and/or CDT for PMVT (page 20), and it is unclear what led to the favorable outcomes in this paper. The authors should emphasize on the strength and specificity of the described technique compared to previous techniques.

2.     Related to the above, it sounds difficult to perform catheterization and stenting in a thrombotic occlusion site without preceding thromlytic therapy. Does this procedure require extensive experience, and were the physicians who performed this procedure experienced?

3.     Additionally, lack of the control group is one of the strong limitations. It may be difficult to set a parallel control group, but is it possible to set a historical control group? If it is inevitably difficult, this should be stated as a limitation.

4.     Because this paper was regarding the observational study/case series, the authors should describe a STROBE flow chart with the number of patients assessed for eligibility, fulfilling inclusion criteria, excluded, refused to participate (maybe opt out method was applied), lost to follow-up, or consequently analyzed.

5.     Although the authors described that this regimen was effective for both acute and chronic PMVT in the Abstract section, however, it is unclear from the description of the Abstract.

6.     Additionally, were there any differences in the outcomes and difficulty, spent time, or the amount of thrombolytic agents in the procedure between acute and chronic PMVT?

Minor concerns and questions for authors:

1.     Page 4 – The authors described that “All patients gave informed consent before the procedure,” however, this is a retrospective study. Were informed consents to participate in this study obtained? Was opt-out method applied?

2.     Page 5 – Please clarify the definition of partial and complete recanalization (PR and CR).

3.     What was the cause of death in one 95-year old patient? Was it associated with the procedure or primary disease?

4.     Sorry for a naïve question, but why this study limited to non-cirrhotic patients?

Author Response

(The authors gave the same response as above.)

Round 2

Reviewer 1 Report

I read with attention corrections and response to the reviewers from Chiang's team: their responses are accurate. The manuscript reading is more confortable and more interesting.

I advise jsut the authors to correct syntax errors in table 3.

Author Response

Thank you for your further comment.

Table 3 had been corrected.

Reviewer 2 Report

All of the issues that I pointed out were adequately revised. I have no additional comments.

Author Response

Thank you.